# Unveiling Neurocognitive Disparities in Encoding and Retrieval between Paper and Digital Tablet-Based Learning

**DOI:** 10.3390/brainsci14010076

**Published:** 2024-01-12

**Authors:** Si-An Lee, Jun-Hwa Hong, Na-Yeon Kim, Hye-Min Min, Ha-Min Yang, Si-Hyeon Lee, Seo-Jin Choi, Jin-Hyuck Park

**Affiliations:** 1Department of ICT Convergence, The Graduate School, Soonchunhyang University, Asan 31538, Republic of Korea; iop5213@naver.com; 2Department of Occupational Therapy, Soonchunhyang University, Asan 31538, Republic of Korea; junhwa921@naver.com (J.-H.H.); skdus318@naver.com (N.-Y.K.); dive1nto@naver.com (H.-M.M.); hamin0106@naver.com (H.-M.Y.); ooa11a@naver.com (S.-H.L.); seoiprn@naver.com (S.-J.C.); 3Department of Occupational Therapy, College of Medical Science, Soonchunhyang University, Asan 31538, Republic of Korea

**Keywords:** brain efficiency, learning, memory, education, fNIRS

## Abstract

The widespread use of mobile devices and laptops has replaced traditional paper-based learning and the question of how the brain efficiency of digital tablet-based learning differs from that of paper-based learning remains unclear. The purpose of this study was to investigate the difference in brain efficiency for learning between paper-based and digital tablet-based learning by measuring activity in the prefrontal cortex (PFC) using functional near-infrared spectroscopy. Thirty-two subjects were randomly assigned to the paper-based learning or the digital tablet-based learning group. Subjects in each group performed a memory task that required memorizing a three-minute novel (encoding phase) on a paper or digital tablet, followed by a test in which they answered four multiple-choice questions based on the novel’s content. To compare both groups, behavioral performance on the test (retrieval phase) and activity in the PFC were measured. As a result, no significant difference in behavioral performance between both groups was observed (*p* > 0.05). However, the paper-based learning group showed significantly lower activity in the PFC in the encoding phase than the digital tablet-based learning group (*p* < 0.05) but not in the retrieval phase. The current study demonstrated that brain efficiency in encoding is higher in subjects with paper-based learning than those with digital tablet-based learning. This finding has important implications for education, particularly in terms of the pros and cons of electronic document-based learning.

## 1. Introduction

In recent times, there has been a growth in the popularity of digital learning facilitated by electronic media, including personal computers (PCs), smartphones, and digital tablets, driven by the widespread distribution of electronic devices for educational purposes [1]. Consequently, electronic documents are gradually replacing their print counterparts. 

This shift in educational media has sparked a certain level of skepticism [2]. However, it is essential to note that previous studies have indicated that this skepticism may be more reflective of a general cultural attitude toward digital learning rather than a measurable cognitive effort during the reading process [2]. Notably, a previous study has suggested that there is insignificant variation in learning outcomes between different learning media. Specifically, electronic documents that optimize hypertext and multimedia elements for student engagement can enhance learning speed and memory recall [3].

On the contrary, some studies have indicated a preference among students for paper-based learning [4], considering information from paper media to be more trustworthy [3]. Furthermore, concerns have been raised regarding the impact of digital learning on deep reading skills, comprehension, and the development of long-term knowledge. Indeed, in a previous study, students were divided into two groups, with one reading 1400 to 2000 words in print and the other reading the same text in PDF format on a computer screen. The result showed significantly higher scores for the group reading on paper, reinforcing concern about the use of electronic media for learning [1,5]. 

These conflicting research findings on digital learning underscore the need for a rational examination of the controversy. One reasonable explanation for this disparity in results is that previous studies mainly measured behavioral outcomes to assess the efficacy of digital learning. Specifically, tests were based on the time spent on learning tasks and accuracy on test items, leading to inconsistent evaluation criteria across studies [3,4,6,7]. Therefore, to compare learning efficacy between learning media, it is crucial to conduct observations not only of behavioral outcomes but also of activity in brain areas relevant to learning. 

To address this issue, previous studies included measurements of brain activity with behavioral outcomes. In a recent study, 38 participants were allocated into learning groups using notebooks, digital tablets, and mobile phones, and brain activity was measured using functional near-infrared spectroscopy (fNIRS) while performing a learning task focused on encoding and recalling scheduled events on a calendar. The findings indicated that accuracy in the retrieval was not significantly different among the three groups. On the other hand, the notebook learning group showed higher brain activity during the memory retrieval phases compared to the other groups [8]. Since task performances were similar among the three groups, higher brain activity in the notebook learning group could be considered using a greater amount of brain energy to produce the same level of results. In other words, paper-based learning was identified to be less efficient for learning. Conversely, another study has shown that paper-based learning was more efficient for learning. In this study, 11 participants were assigned to the paper-based or digital table-based learning group for a sentence reading task, and their brain activity was measured. The task involved memorizing underlined words while reading sentences, followed by a test. Although test performances did not significantly differ between the two groups, the digital tablet-based learning group showed higher brain activity while performing the task [1]. Furthermore, despite the fact that the stages of memory are divided into encoding and retrieval, and it has been shown that the brain offers different contributions in the encoding and retrieval phases [9,10,11], previous studies have not analyzed brain efficiency by learning media by stages, making it difficult to clearly identify at which stage there is a difference in brain efficiency [1,8]

Consequently, there is ongoing debate regarding brain efficiency associated with learning through different media, and the question of how the brain efficiency of digital tablet-based learning differs from that of paper-based learning remains unclear. Furthermore, neither of the studies presented brain activity for the encoding and retrieval phases, leading to a challenge in pinpointing the specific phase where differences in brain efficiency occur. Therefore, the objective of this study was to investigate the brain efficiency between paper-based and digital tablet-based learning by measuring activity in brain areas related to learning using fNIRS, with the assessment conducted by dividing brain activity into encoding and retrieval phases. This study specifically focused on working memory, a crucial factor in learning, and measured the activity in the prefrontal cortex (PFC) while performing a memory task. 

## 2. Materials and Methods

### 2.1. Procedure

This study investigated the effects of paper-based and digital tablet-based learning on brain activity by conducting an experiment using fNIRS while subjects engaged in a brief memory task. Subjects were randomly assigned into either the paper-based learning group or the digital tablet-based learning group, with each group completing the memory task on an allocated medium. Changes in oxygenated hemoglobin (HbO) concentration in the PFC were measured during the memory task. The experiment was separately implemented for each subject with one experimenter. The completion of the task for both media required 3 min. 

### 2.2. Participants

Thirty-two healthy subjects in their twenties (twenty-four females) were recruited to participate in this study. Eligibility criteria were as follows: (1) no vision deficiency or mental illness, (2) not undergoing pharmacological or non-pharmacological treatments for cognitive function, (3) intact comprehension of verbal instructions, and (4) at least six months of experience learning on a digital tablet. Exclusion criteria were as follows: (1) presence of neurologic or psychiatric disorders, (2) presence of familiarity with the content of the memory task, and (3) unable to read Korean.

All subjects were given explanations before performing the task, and informed consent was obtained from all subjects involved in this study, according to the Declaration of Helsinki. This study was approved by the Institutional Review Board of Soonchunhyang University (202308-SB-089). 

### 2.3. Memory Task

In this study, excerpts from a Japanese novel titled “Suzume’s Gate” were employed as a memory task. The selection of this novel was based on its novelty at the time of the study, and it was anticipated that most subjects would not have encountered it previously. The extracted sentences spanned three pages on A4 paper. Subjects were instructed to read the entire sentences within three minutes, and after the reading, they were provided with four multiple-choice questions to choose the correct one out of four options that could only be solved by recalling the content of the novel.

Subjects in the paper-based learning group received three sheets of A4 paper arranged in the order of the novel’s contents and proceeded to read it. Following the completion of the first sheet, they had to turn over the paper to read the subsequent contents one by one. Subjects in the digital tablet-based learning group read the novel in PDF format on a 10.9-inch tablet (iPad, Apple Inc., Cupertino, CA, USA). Throughout the reading phase, they refrained from interacting with the screen to zoom in or out. After reading the first sheet, they scrolled up and down the screen to read the remaining contents sequentially.

Upon completion of the reading task in each medium, subjects in both groups were instantly provided with a sheet of A4 paper containing four multiple-choice questions and a ballpoint pen to choose the correct answer. The activity in the PFC of subjects was measured using fNIRS throughout the entire encoding and retrieval phases, including a 30-s baseline before each phase.

### 2.4. Measurement

A multi-channel fNIRS device (OctaMon, Artinis Medical Systems BC, Elst, The Netherlands) with 8 light sources and 2 lighter detectors was used to measure the activity in the PFC at a sampling rate of 10 Hz. A total of 8 channels were located throughout the PFC (4 channels: right PFC and 4 channels: left PFC) following the international 10–20 electroencephalography (EEG) placement system [12,13] (Figure 1). The wavelengths used for oxygenated hemoglobin (HbO) and deoxygenated hemoglobin (HbR) detection were 760 and 850 nm, respectively. All light sources and detectors were mounted to an elastic headband, which ensured that light sources and detectors made good contact with the subject’s forehead.

### 2.5. Data Preprocessing

The fNIRS data were collected using OxySoft software (version 3.0.52. Artinis Medical Systems BV, Elst, The Netherlands). This study only utilized HbO concentrations, which are more sensitive than HbR to cognitive task-related changes [14], from the PFC and they were averaged. Each fNIRS channel was visually inspected and channels with large spikes with a standard deviation of 300 μM/mm from the mean, which is considered noisy, were excluded from analysis [13]. 

The fNIRS data were preprocessed by a 4th-order Butterworth band-pass filter (cut-off 0.01–0.2 Hz) and an eigenvector-based spatial filtering [13] to reduce artifacts including respiration and cardiac interference. The concentration changes of hemoglobin were computed according to the modified Beer–Lambert Law.

For each cleaned fNIRS channel, the HbO concentrations of the encoding and retrieval phases were segmented. Subsequently, all segmented HbO concentrations were averaged for each channel and each subject. The segments were normalized by subtracting the mean value of a previous baseline from the mean in each segment to remove the intra-individual variance of the starting value.

### 2.6. Statistical Analysis

All data were analyzed by using SPSS version 22.0. The Kolmogrov–Smirov test confirmed the normality of data, so parametric statistics were used in this study. The graphical exploration of the data was conducted by box plots. To examine the differences in demographic characteristics of subjects between both groups, the Chi-square or independent *t*-test was used. To compare the performance on the memory task and HbO concentration during the encoding and retrieval phases between both groups, the independent *t*-test was used. All statistical significances were set at *p* < 0.05.

## 3. Results

### 3.1. Demographic Characteristics

Table 1 shows the demographic characteristics of the subjects. Both groups were dominated by females, with more than half of the subjects. The mean ages of the paper-based and digital tablet-based learning groups were 22.19 and 21.56, respectively. There was no significant difference in sex ratio and ages between both groups (*p*’s > 0.05) (Table 1). 

### 3.2. Behavioral Performance and Brain Activity

There was no significant difference in memory performance between both groups (*p* > 0.05) (Table 2) (Figure 2). On the other hand, there was a significant difference in the mean of HbO concentration during the encoding phase between both groups (*p* < 0.05) (Table 2). Specifically, the paper-based learning group showed lower activity in the PFC than the digital tablet-based learning group. However, the mean of HbO concentration during the retrieval phase did not significantly differ (Table 2) (Figure 2). 

### 3.3. Correlation in Behavioral Performance with Hemodynamic Response

Behavioral performance was not found to be significantly correlated with HbO concentration during the encoding phase (*p* > 0.05) and the retrieval phase (*p* > 0.05) (Table 3). This finding suggests that the observed PFC activity during the encoding and retrieval phase did not exhibit a discernible association with memory accuracy.

## 4. Discussion

Using two groups of subjects who performed a simple memory task using paper or a digital tablet, followed by a retrieval task, this study confirmed two major findings. First, accuracy was not significantly different between the paper-based learning and digital tablet-based learning groups. Second, activity in the PFC during the encoding phase was significantly lower in the paper-based learning group than the digital tablet-based learning group, but not during the retrieval phase. In other words, subjects in the paper-based learning group used less prefrontal energy during the encoding phase to achieve the same level of performance as the digital tablet-based learning group. These findings suggest that the use of paper could lead to higher brain efficiency in encoding information, which is consistent with the findings of a previous study [1].

The channels from the fNIRS device covered the PFC, which is known to be correlated with working memory and memory encoding [15]. During the encoding phase, the PFC in the digital tablet-based learning group was more activated than the paper-based learning group. This suggests that subjects in the digital tablet-based learning group might experience a higher mental workload than the paper-based learning group. In a previous study, the display screen of tablet media increases the degree of mental fatigue and burden when performing mental tasks compared to printed media [16], supporting the findings of this study. This is an interesting result given that this study only included subjects who already had enough experience using digital tablets for learning, which means that they are comfortable using them but are using their brain inefficiently when it comes to encoding. 

This difference in brain efficiency based on a learning medium could be explained by spatial attributes of learning information. In this study, subjects in both groups performed a visual memory task, where the brain recalled information in relation to its context [17]. In other words, one can remember something specifically by its location within a medium. During the encoding phase, subjects in the paper-based learning group read text fixed on each paper, while subjects in the tablet-based learning group had to scroll up and down the screen of a digital tablet, so the text did not remain fixed in the same place on the screen, but changed positions [5,18]. Indeed, in a previous study, having to scroll on a screen could make remembering more difficult and inefficient [18], supporting the findings of the current study. Taken together, the aid of spatial attributes for paper-based learning might lead to higher brain efficiency. 

There is a growing body of evidence supporting the superiority of paper-based learning. A previous study has indicated that the transition of knowledge from episodic memory to semantic memory was more efficient from paper than a screen, even though minimal speed and recall differences were observed [19]. Another previous study has reported that students’ comprehension of information was greater from paper than from electronic documents [7]. In recent studies, the behavioral performance of a memory task was almost the same for paper and table media but activity in the brain while subjects performed the task on the digital tablet was higher than the paper medium [1,8], which is in line with the findings of this study. 

On the other hand, there was no considerable difference in activity in the PFC during the retrieval phase. The neural mechanism of the PFC in memory could be one possible explanation for this finding. The PFC is crucially involved not only in memory encoding but also in the memory retrieval process [20]. However, the multiple-choice questions used in this study could be considered to assess recognition rather than pure recall as the contents of the answer option could be a cue for recall. Neuroimaging studies have consistently reported that the PFC is not necessary for recognition [20,21]. Instead, the medial temporal lobes are heavily involved in the recognition [22], supporting that there was no significant difference in activity in the PFC between the two learning media during the retrieval phase. Indeed, in a previous study, a significant difference between the paper and digital tablet-based learning groups was found in the hippocampus during the retrieval phase using multiple-choice questions to assess memory [21]. These findings suggest that there might be different brain regions to observe depending on the specific methods of retrieval.

On the other hand, several studies have demonstrated that electronic documents that optimize hypertext and multimedia to engage students could lead to improved outcomes [3,6,22], refuting the notion that paper-based learning is unconditionally efficient for learning. In a previous study, students using electronic documents were more actively engaged in learning than printed media [6]. In another study, the ability of hypertext to integrate information within electronic documents positively affects cognitive processing, resulting in a deeper understanding of the content [23]. Taken together, table PC-based learning designed for active learning could be efficient for learning. 

Contrary to previous related works, this study investigated brain efficiency for learning using encoding and retrieval phases. Previous studies have not distinguished brain efficiency between both phases, making it unclear at which point brain efficiency is higher, whereas the current findings suggest that paper-based learning might ensure higher brain efficiency during the encoding phase [1,8]. Furthermore, the significance of this study is that it confirmed the effects of the use of media on brain efficiency in learning by eliminating other learning factors that might affect learning, such as the use of a pen or stylus [8]. 

Even though this is the first study to identify brain efficiency by memory phases, there are some limitations. Firstly, this study could not compare brain efficiency across mediums for different forms of learning. While this study investigated a passive form of learning, such as simply reading a given text, a comparison of brain efficiency for active learning, such as annotating, taking notes, or underlining, might yield different results [8]. Secondly, this study could not determine brain efficiency for recall by using only multiple-choice questions in the retrieval phase, limiting the conclusion that there is no difference in brain efficiency between paper- and digital tablet-based learning in the retrieval. Thirdly, the lack of a standardized paradigm (block or event paradigm) hinders direct comparisons with the existing literature that often employs these methodologies [24,25]. To overcome these limitations, future research will adopt a block design or event paradigm in fNIRS experiments, ensuring more meaningful comparisons. Fourthly, given that this study only includes young adults in their 20s, and all of them were college students, the generalizability of the current findings is limited. Finally, due to a lack of channels on the fNIRS device, other brain regions related to learning were not measured. Although the PFC is mainly involved in encoding and retrieval, temporal lobes are also highly correlated with encoding and retrieval [26]. Future research will need to compare brain efficiency differences across learning mediums for a variety of types of learning.

## 5. Conclusions

The current study demonstrated that brain efficiency in encoding is higher in subjects undergoing paper-based learning than those undergoing tablet-based learning. This finding could have important implications for education, particularly in terms of the pros and cons of electronic document-based learning. The widespread use of mobile devices and laptops has replaced traditional paper-based learning, which can put a strain on the brain when it comes to encoding. Further studies are needed to determine how learning with electronic documents over a longer period can change brain efficiency for learning.

## Figures and Tables

**Figure 1 brainsci-14-00076-f001:**
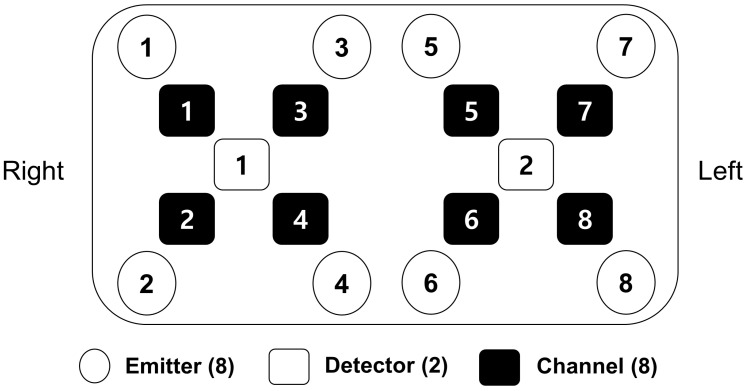
The placement of eight emitters, two detectors, and eight channels.

**Figure 2 brainsci-14-00076-f002:**
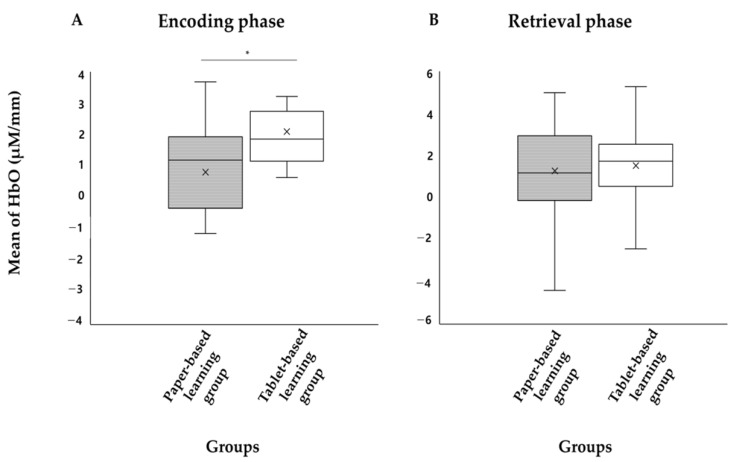
Boxplots of the effect of learning media on encoding (**A**) and retrieval phase (**B**). * *p* < 0.05.

**Table 1 brainsci-14-00076-t001:** Demographic characteristics of subjects in each group.

Demographic Characteristics		Paper-Based Learning Group (*n* = 16)	Digital Tablet-Based Learning Group (*n* = 16)	χ^2^/t
Sex	Male	5 (31.3%)	2 (12.5%)	1.646
	Female	11 (68.7%)	14 (87.5%)	
Age (years)		22.19 ± 2.10	21.56 ± 1.15	1.042

**Table 2 brainsci-14-00076-t002:** Comparison of behavioral performance and brain activity between both groups.

Characteristics		Paper-Based Learning Group (*n* = 16)	Digital Tablet-Based Learning Group (*n* = 16)	t
Memory task performance (counts)		3.63 ± 0.50	3.75 ± 0.44	0.745
Mean of HbO (μM/mm)	Encoding	0.811 ± 2.101	2.105 ± 1.777	2.148 *
	Retrieval	1.218 ± 2.359	1.532 ± 1.741	0.347

* *p* < 0.05.

**Table 3 brainsci-14-00076-t003:** Correlation of behavioral performance with hemodynamic response.

Characteristics	Mean of HbO (μM/mm)	
	Encoding Phase	Retrieval Phase
Memory task performance (counts)	0.270	0.340

## Data Availability

The group data presented in this study are available upon request from the corresponding author. The individual data are not publicly available due to privacy and confidentiality.

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
