# Peer review of "Unveiling Neurocognitive Disparities in Encoding and Retrieval between Paper and Digital Tablet-Based Learning"

_brainsci, 2024, doi:10.3390/brainsci14010076_

Round 1
Reviewer 1 Report
Comments and Suggestions for Authors
The paper reports of the comparison between the brain efficiency associated to paper- and Tablet-based learning during encoding and retrieval phases. The topic is of great interest and the manuscript is well written. In my opinion, few concerns related to the fNIRS data analysis need to be addressed:
1) It is not clear to me how the mean value of the HbO was computed. In fact, The Octamon system is a continuous wave system, hence, it could be necessary to normalize the mean value of the experimental phase with respect to a previous baseline. Please specify this aspect in the manuscript.
2) It seems that all the channels were averaged. In this case, maybe, some spatial information was lost. In my opinion, results for each channel should be reported in order to investigate also differences in the left or right hemispheres. In this case, a multiple comparison correction should be added.
3) It could be useful to report an example of the fNIRS signal for both the experimental groups, and also boxplots summarizing the metrics considered in the statistical analysis.
4) In my opinion, since the protocol is not standard (i.e., event or block paradigm), some particular data analysis, together with the mean value of the HBO, could be applied. In fact, in the literature some approaches based on the complexity of the fNIRS signals or specific algorithms developed to detect brain activity also in ecological situations have been proposed. Please, consider adding these analyses in the manuscript, or, at least, to describe in the Discussion section these methodologies as a further development of the study. Please refer to:
· Perpetuini, D., Chiarelli, A. M., Cardone, D., Filippini, C., Bucco, R., Zito, M., & Merla, A. (2019). Complexity of frontal cortex fNIRS can support Alzheimer disease diagnosis in memory and visuo-spatial tests. Entropy, 21(1), 26.
· Pinti, P., Merla, A., Aichelburg, C., Lind, F., Power, S., Swingler, E., ... & Tachtsidis, I. (2017). A novel GLM-based method for the Automatic IDentification of functional Events (AIDE) in fNIRS data recorded in naturalistic environments. Neuroimage, 155, 291-304.
Author Response
We sincerely appreciate your constructive review of our paper.
We have attached our response to your comments.

Reviewer 2 Report
Comments and Suggestions for Authors
Traditional paper-based learning has been supplanted by the ubiquitous use of laptops and mobile devices, and it is yet unknown how the brain efficiency of learning on a tablet PC differs from studying on paper. The goal of this study was to compare the brain efficiency for learning between studying on paper and learning on a tablet PC by monitoring prefrontal cortex (PFC) activity using functional near-infrared spectroscopy. 32 participants were randomized at random to either the tablet PC-based learning group or the paper-based learning group. Participants in each group completed a memory test consisting of four multiple-choice questions based on the content of a three-minute book that they had to memorize on paper or on a tablet PC.
Behavioral performance on the test (retrieval phase) and PFC activity were evaluated to compare the two groups. Consequently, there was no discernible variation in the behavioral performance of the two groups (p > 0.05). In contrast to the tablet PC-based learning group, the paper-based learning group demonstrated significantly less activity in the PFC throughout the encoding phase (p < 0.05), but not during the retrieval phase. The results of this study showed that people who learned using a tablet PC had lower brain encoding efficiency than those who learned using paper. This discovery has significant educational ramifications, especially when considering the benefits and drawbacks of electronic document-based learning.
- The Title is misleading. It does not provide the insights of the paper. The authors must modify the title.
- The term PC tablet is not suitable. Please use some better wording to replace it.
- Which statistical tests were applied. Why the authors have used non-parametric statistical test. Please clarify.
- The literature review is incomplete. Several fNIRS studies based on such tasks were done and not acknowledged. There should be the enough research gap which warants the current study.
- how the number of subjects were selected? What is the basis of utilizing less number of subjects.
- Table 2 is not understandable. Which statistical test was reported there?
- Why there is no figure showing the hemodyanmics response with sTD. Strange to see the results section. Very brief.
- It is not clear whether the behavioural results were related with fNIRS? Please justify.
Author Response
We sincerely appreciate your constructive review of our paper.
We have attached our response to your comment.

Round 2
Reviewer 1 Report
Comments and Suggestions for Authors
I thank the Authors for addressing all my concerns. In my opinion the paper is improved and suitable for publication in the present form.
Reviewer 2 Report
Comments and Suggestions for Authors
The authors have addressed my coments and the manuscript is now in good shape of publication.